# How Contaminated Is Your Benchmark?
# Measuring Dataset Leakage in Large Language Models with Kernel Divergence

Hyeong Kyu Choi [* 1]   Maxim Khanov [* 1]   Hongxin Wei [2]   Yixuan Li [1]

## Abstract

Dataset contamination, where evaluation datasets overlap with pre-training corpora, inflates performance metrics and undermines the reliability of model evaluations. Measuring dataset contamination thus becomes essential to ensure that performance evaluations genuinely reflect a model's ability to generalize to unseen data, rather than relying on memorized examples. To address this problem, we propose Kernel Divergence Score (KDS), a novel method that evaluates dataset contamination by computing the divergence between the kernel similarity matrix of sample embeddings, before and after fine-tuning on the benchmark dataset. Leveraging the insight that fine-tuning affects unseen samples more significantly than seen ones, KDS provides a reliable measure of contamination. Through extensive experiments on controlled contamination scenarios, KDS demonstrates a near-perfect correlation with contamination levels and outperforms existing baselines. Additionally, we perform comprehensive ablation studies to analyze the impact of key design choices, providing deeper insights into the components and effectiveness of KDS. These ablations highlight the importance of leveraging fine-grained kernel-based information and confirm the reliability of the proposed framework across diverse datasets and settings. Code is released in https://github.com/deeplearning-wisc/kernel-divergence-score.

*Equal contribution   [1]Department of Computer Sciences, University of Wisconsin–Madison, United States [2]Department of Statistics and Data Science, Southern University of Science and Technology, China. Correspondence to: Yixuan Li <sharonli@cs.wisc.edu>.

*Proceedings of the 42nd International Conference on Machine Learning*, Vancouver, Canada. PMLR 267, 2025. Copyright 2025 by the author(s).

## 1. Introduction

When a large language model (LLM) performs remarkably well on a benchmark, can we confidently attribute its success to true generalization—or is it simply a reflection of what the model has already seen during pre-training? The reality is, we often don't know. Beneath the surface of those impressive performance scores lies a critical vulnerability: *dataset contamination*, a phenomenon where evaluation datasets overlap with the pretraining data of the model (Golchin & Surdeanu, 2024). This overlap artificially inflates reported performance metrics, obscures true generalization capabilities, and raises critical concerns about the reliability of benchmark evaluations. This brings us to a pressing and underexplored question: ***How can we measure the degree of dataset contamination?***

Addressing this question is crucial to ensuring that performance evaluations genuinely reflect a model's ability to generalize to unseen data, rather than benefiting from overlap with pretraining data. To formalize the problem, we aim to develop a scoring function $S : (\mathcal{D}, \mathcal{M}) \to \mathbb{R}$, that takes a benchmark dataset $\mathcal{D}$ as input and produces a score indicative of its relative contamination level with respect to the given model $\mathcal{M}$. A higher score corresponds to a greater contamination level. Such a score is valuable because researchers can use it to rank multiple benchmarks and prioritize the less contaminated ones, enabling more informed comparisons and reliable evaluation. For the score to be reliable, we argue that the scoring function must satisfy two essential properties: *monotonicity*, which ensures that the score exhibits a positive correlation with the contamination level, and *consistency*, which means that the score remains stable across independently sampled subsets with the same contamination rate.

To measure dataset contamination, we introduce the **Kernel Divergence Score** (KDS), which computes the divergence of the kernel similarity matrix of sample embeddings before and after fine-tuning on the benchmark dataset. By analyzing changes in the kernel similarity matrix, KDS captures how fine-tuning reshapes the embeddings for seen and unseen data, providing a more holistic and nuanced perspective on dataset contamination. This approach is motivated by the fact that fine-tuning has a more significant effect on the

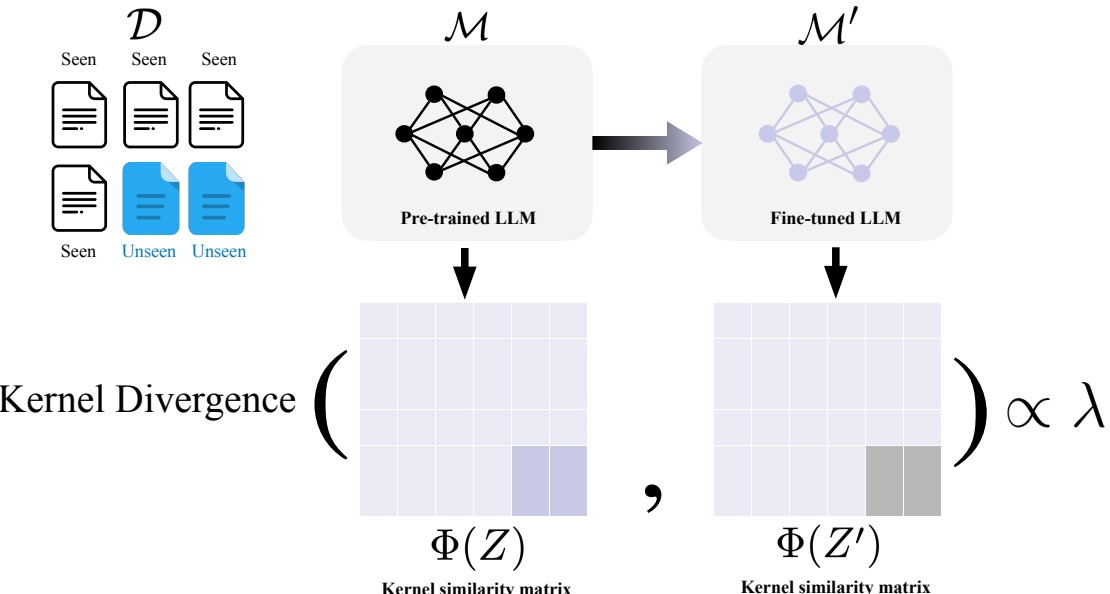

*Figure 1.* Overview of the proposed **Kernel Divergence Score** (KDS) framework for measuring dataset contamination in large language models. The process involves extracting sample embeddings from the model before and after fine-tuning on the benchmark dataset $\mathcal{D}$, computing the kernel similarity matrix for each stage, and measuring the divergence between the two matrices $\Phi(Z)$ and $\Phi(Z')$. By capturing the changes of embeddings induced by fine-tuning, KDS provides a reliable and interpretable score to evaluate the level of dataset contamination.

embeddings of unseen samples, which the model must adapt to, while seen samples exhibit minimal changes due to prior exposure during pre-training. Furthermore, as the proportion of unseen samples increases, their cumulative effect on the kernel divergence score becomes more pronounced. By evaluating these changes, KDS can provide a reliable and interpretable measure of dataset contamination, with scores that proportionally reflect the level of contamination.

To evaluate KDS, we perform extensive experiments, systematically controlling contamination ratios across multiple datasets. Our results demonstrate that KDS achieves near-perfect correlation with contamination levels, generally outperforming existing baselines across multiple datasets. Additionally, we show that KDS is robust to design choices, including kernel functions, kernel bandwidth, and the extraction location of embeddings. Overall, KDS provides stable scores across diverse scenarios, enabling researchers to reliably identify benchmarks based on contamination levels. We summarize our contributions as follows:

- We propose *Kernel Divergence Score*, a reliable dataset-level scoring function to measure benchmark contamination. To the best of our knowledge, we are the first to leverage the fine-grained information of kernels for scoring contamination levels.

- We validate Kernel Divergence Score through extensive experiments on controlled contamination scenarios, showing strong performance over existing baselines.

- We perform comprehensive ablations to analyze the impact of various design choices. Further practical discussions are presented, providing deeper insights into our kernel-based approach.

## 2. Problem Statement

### 2.1. Measuring Benchmark Contamination

The objective is to evaluate the relative degree to which a benchmark evaluation dataset, $\mathcal{D}$, has been exposed to the pre-training process of a given LLM, $\mathcal{M}$. In modern LLMs, the pre-training dataset is typically unavailable, making it difficult to directly assess the contamination level. Accordingly, we consider a generalized characterization of the benchmark evaluation data, modeling it as a mixture of both seen and unseen data:

$$\begin{aligned} \mathcal{D} &= \mathcal{D}_{\mathcal{M}}^{\text{seen}} \cup \mathcal{D}_{\mathcal{M}}^{\text{unseen}} \\ |\mathcal{D}_{\mathcal{M}}^{\text{seen}}|/|\mathcal{D}| &= \lambda, \end{aligned} \quad (1)$$

where $\mathcal{D}_{\mathcal{M}}^{\text{seen}}$ is the data seen during $\mathcal{M}$'s pre-training, $\mathcal{D}_{\mathcal{M}}^{\text{unseen}}$ is the data not seen by $\mathcal{M}$, and $\lambda \in [0,1]$ is an unknown

parameter indicating the fraction of seen data in $\mathcal{D}$. Within this framework, we aim to develop a dataset-level scoring function

$$S : (\mathcal{D}, \mathcal{M}) \to \mathbb{R},$$

which measures the relative level of contamination of dataset $\mathcal{D}$ with respect to model $\mathcal{M}$. A larger score indicates more contamination and vice versa.

A reliable scoring function is practically valuable because it allows us to identify datasets that are contaminated with respect to model $\mathcal{M}$. Hence, we now outline the desirable properties that the scoring function $S$ should satisfy.

### 2.2. Reliable Contamination Scores

In this section, we state two essential requirements for a reliable contamination scoring function: **Monotonicity** and **Consistency**.

**Requirement 1.** *(Monotonicity) If dataset $\mathcal{D}$ is more independent of model $\mathcal{M}$ than dataset $\mathcal{D}'$, i.e., $\lambda < \lambda'$, then*

$$S(\mathcal{D}, \mathcal{M}) < S(\mathcal{D}', \mathcal{M})$$

*should hold with statistical significance. In other words, a dataset with a smaller $\lambda$, the fraction of seen data, should have accordingly a smaller contamination score $S(\mathcal{D}, \mathcal{M})$.*

**Requirement 2.** *(Consistency) If datasets $\mathcal{D}$ and $\mathcal{D}'$ both comprise of independently and identically distributed (i.i.d.) samples from a distribution with the same contamination ratio $\lambda$,*

$$S(\mathcal{D}, \mathcal{M}) \approx S(\mathcal{D}', \mathcal{M})$$

*should hold with statistical significance.*

The Monotonicity requirement ensures that the scoring function exhibits a **positive correlation** with the dataset's contamination rate, even though the true contamination rate is typically unknown in real-world scenarios. A scoring function satisfying this requirement enables reliable ranking of datasets for each model based on their contamination scores. The Consistency requirement, on the other hand, ensures that the scores are robust to variations in the specific samples drawn from the same underlying distribution, under the same $\lambda$. This property ensures that the randomness induced from sampling does not substantially affect the overall scoring.

## 3. Method: Kernel Divergence Score

In this section, we present our method, Kernel Divergence Score, which leverages information among samples within the model's embedding space to establish a more nuanced contamination scoring mechanism. In a nutshell, we assess changes in the kernel matrix of sample embeddings before and after fine-tuning, capturing how the relationships between samples evolve as a result of fine-tuning.

Our approach is motivated by the fact that *fine-tuning affects the embedding relationships involving unseen samples more significantly than those involving seen samples*. For seen samples, the model has already been exposed to similar data during pretraining, leading to minimal shifts in their embedding relationships. In contrast, unseen samples experience more pronounced changes, as the fine-tuning process adjusts the model to better align with the benchmark dataset. By evaluating these changes using the Kernel Divergence Score, we can provide a reliable and granular measure of dataset contamination.

**Kernel similarity matrix.** A kernel similarity matrix captures the relationships among data samples, providing fine-grained information on their distribution. Formally, let $Z \in \mathbb{R}^{n \times d}$ represent the embeddings of $n$ samples in the dataset $\mathcal{D}$, where $Z_i \in \mathbb{R}^{1 \times d}$ is the normalized embedding of $i$-th sample extracted from the pre-trained LLM $\mathcal{M}$. We define the kernel matrix $\Phi(Z) \in \mathbb{R}^{n \times n}$ based on the Radial Basis Function (RBF) kernel:

$$\Phi(Z)_{i,j} = \exp(-\gamma ||Z_i - Z_j||_2^2),$$

where $\Phi(Z)_{i,j}$ is the kernel similarity between samples $Z_i$ and $Z_j$, and $\gamma$ controls the kernel bandwidth. The kernel matrix captures the pairwise relationships between all samples in the dataset, with values ranging from 0 to 1. An entry close to 1 indicates high similarity (small distance), while a value close to 0 indicates low similarity (large distance). The matrix $\Phi(Z)$ is both symmetric and positive semidefinite.

**Leveraging the effect of fine-tuning.** Our proposed Kernel Divergence Score is based on the kernel matrix before and after fine-tuning. Formally, let $Z' \in \mathbb{R}^{n \times d}$ represent the embeddings *after* LoRA (Hu et al., 2022) fine-tuning on dataset $\mathcal{D}$, where $Z'_i \in \mathbb{R}^{1 \times d}$. Accordingly, we can derive the kernel similarity matrix as:

$$\Phi(Z')_{i,j} = \exp(-\gamma ||Z'_i - Z'_j||_2^2). \tag{2}$$

Then, we define **Kernel Divergence** as

$$\frac{1}{E} \sum_{i,j=1}^{n} \left| \Phi(Z)_{i,j} \log \frac{\Phi(Z)_{i,j}}{\Phi(Z')_{i,j}} \right|, \tag{3}$$

where $E = \sqrt{\sum_{i,j} \Phi(Z)_{i,j}}$ is a normalizer. When $\gamma = 1$, our score in Eq. (3) can be equivalently written as

$$\frac{1}{E} \sum_{i,j=1}^{n} \underbrace{\exp(-||Z_i - Z_j||_2^2)}_{\substack{(1) \text{ Soft gating for originally} \\ \text{closely-related samples}}} \underbrace{\left| ||Z'_i - Z'_j||_2^2 - ||Z_i - Z_j||_2^2 \right|}_{(2) \text{ Change in distance before and after SFT}} . \tag{4}$$

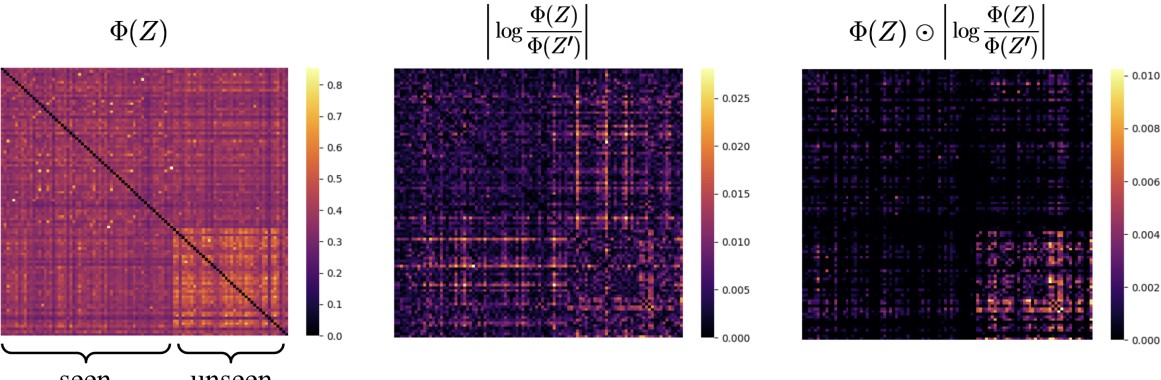

*Figure 2.* **Decomposition of the Kernel Divergence Score.** Each component of the Kernel Divergence Score function is shown. $\Phi(\cdot)$ denotes the kernel similarity matrix, $Z$ and $Z'$ represent normalized sample embeddings before and after fine-tuning, and $\odot$ is the Hadamard product. Score and embeddings are based on Llama-3.1-8B-Instruct (Dubey et al., 2024). (**Left**) shows that the original kernel similarity matrix before fine-tuning. Note, that diagonal values are zeroed for better visualization, because all diagonal values are 1 in RBF kernels. (**Middle**) reveals that fine-tuning alters relationships among unseen samples more than those among seen samples. (**Right**) Combining the two panels enhances the distinction between seen and unseen samples, thereby enabling a more reliable measurement of contamination levels.

**Interpretation of kernel divergence.** This function quantifies how fine-tuning changes the pairwise distances between samples, weighted by their original similarity. Specifically, the second term $\left| ||Z'_i - Z'_j||_2^2 - ||Z_i - Z_j||_2^2 \right|$ measures the absolute change in the squared Euclidean distance between embedding pairs caused by fine-tuning. For unseen samples, fine-tuning tends to create new meaningful relationships or significantly alter their embeddings, making their contribution to the score larger. The first exponential term acts as a soft gating function, assigning a higher weight to pairs of samples that were originally closer to each other. By incorporating this term, the score prioritizes the impact of fine-tuning on pairs that were initially similar, highlighting cases where fine-tuning induces significant changes in their relationships. Overall, a larger fraction of unseen examples (or smaller $\lambda$) can elevate the kernel divergence more significantly. Because we want the scoring function $S(\mathcal{D}, \mathcal{M})$ to be positively correlated with the contamination rate $\lambda$, we define our final scoring function to be the negation of kernel divergence:

$$S(\mathcal{D}, \mathcal{M}) = -\frac{1}{E} \sum_{i,j=1}^{n} \left| \Phi(Z)_{i,j} \log \frac{\Phi(Z)_{i,j}}{\Phi(Z')_{i,j}} \right|, \quad (5)$$

which is expected to be larger as the contamination $\lambda$ grows.

**Visual demonstration of score components.** To provide a concrete understanding of the role of each component in our Kernel Divergence Score (Eq. (3) or Eq. (4)), we present the kernel matrix for each component in Figure 2. We use $n = 100$ samples with a contamination rate $\lambda = 0.4$, meaning 40% of the samples are seen during model pre-training. The

left panel illustrates the soft gate from Eq. (4), or the kernel similarity matrix using the pre-trained embedding. The middle panel captures the change of pairwise embedding distance after supervised fine-tuning, and the right panel shows the resulting element-wise score matrix after the Hadamard product of the two. As hypothesized, the middle panel suggests that relationships involving unseen samples are more significantly altered by fine-tuning. By multiplying the soft gate (left panel), unseen sample pairs contribute more to the overall score. Additional examples with varying contamination rates are provided in Appendix B.

## 4. Experiments

In this section, we conduct a controlled experiment to verify the reliability of our Kernel Divergence Score with respect to the Monotonicity and Consistency requirements (*c.f.* Section 2.2). For a comprehensive analysis, we also assess existing non-kernel-based scoring methods.

### 4.1. Experimental Setup

**Dataset and model.** Our controlled experiment aims to evaluate the extent to which each method satisfies the Monotonicity and Consistency requirements. For this purpose, we utilize three popular pre-training data detection benchmarks, WikiMIA (Shi et al., 2023), BookMIA (Shi et al., 2023), and ArxivTection (Duarte et al., 2024), each comprising samples labeled as 'seen' or 'unseen'. Among the models compatible with the datasets, we select Mistral-7B-Instruct-v0.2 (Jiang et al., 2023) for our main evaluation. We present additional results for other model families in Section 6.

*Table 1.* **Monotonicity Evaluation.** Correlation coefficients averaged across five different subsets are shown. SRCT on BookMIA and ArxivTection are omitted due to excessive computation. On average, our Kernel Divergence Score demonstrates the best compliance with the Monotonicity requirement.

| Methods | WikiMIA | | BookMIA | | ArxivTection | | *Average* | |
|---|---|---|---|---|---|---|---|---|
| | Spearman ↑ | Pearson ↑ | Spearman ↑ | Pearson ↑ | Spearman ↑ | Pearson ↑ | Spearman ↑ | Pearson ↑ |
| *Non-kernel-based Methods* | | | | | | | | |
| Zlib (Carlini et al., 2021) | 0.968 | 0.960 | -1.000 | -0.997 | 0.997 | 0.918 | 0.322 | 0.294 |
| Zlib + FSD (Zhang et al., 2025) | 0.976 | 0.966 | -0.888 | -0.895 | 0.941 | 0.947 | 0.343 | 0.339 |
| Perplexity (Li, 2023) | 0.933 | 0.929 | 0.964 | 0.967 | **1.000** | 0.997 | 0.966 | 0.964 |
| Perplexity + FSD (Zhang et al., 2025) | 0.979 | 0.967 | -0.777 | -0.824 | 0.992 | 0.982 | 0.398 | 0.375 |
| Min-K% (Shi et al., 2023) | 0.893 | 0.899 | **0.998** | **0.992** | **1.000** | **0.998** | 0.964 | 0.964 |
| Min-K% + FSD (Zhang et al., 2025) | 0.932 | 0.937 | -0.526 | -0.640 | 0.988 | 0.980 | 0.459 | 0.420 |
| Min-K%++ (Zhang et al., 2024b) | -0.790 | -0.833 | OOM | OOM | 0.996 | 0.996 | 0.103 | 0.082 |
| Min-K%++ + FSD (Zhang et al., 2025) | -0.790 | -0.834 | OOM | OOM | 0.754 | 0.809 | -0.018 | -0.013 |
| SRCT (Oren et al., 2024) | 0.080 | 0.073 | - | - | - | - | 0.080 | 0.073 |
| *Kernel-based Method* | | | | | | | | |
| **Kernel Divergence Score** (Ours) | **0.999** | **0.993** | 0.997 | 0.979 | 0.975 | 0.974 | **0.990** | **0.982** |

**Baselines.** We consider various baseline methods for comprehensive evaluation. Specifically, we consider Zlib (Carlini et al., 2021), Perplexity Score (Li, 2023), Min-K% (Shi et al., 2023), Min-K%++ (Zhang et al., 2024b), Fine-tuned Score Deviation (FSD; Zhang et al. (2025)), which evaluate the likelihood of exposure for every sample independently. The overall contamination score of the dataset $S(\mathcal{D}, \mathcal{M})$ is then quantified by averaging these instance-wise scores. In addition, we consider the dataset-level approach that assesses the contamination of a dataset as a whole by examining statistical or distributional patterns that differentiate seen vs unseen datasets. This approach provides a more holistic view of contamination, capturing aggregate characteristics that are not discernible at the individual example level. Specifically, we consider the Sharded Rank Comparison Test (SRCT; Oren et al. (2024)), the latest dataset-level detection method that identifies datasets showing significant variability in likelihood values across different sample orderings. For each baseline, we adjusted their score sign so that higher scores indicate more contamination (*i.e.*, bigger $\lambda$). We include the detailed definition of each baseline in Appendix A.2.

**Experimental details.** For each dataset, we evaluate scoring performances on different contamination rates. For integrity across experiments, we fix the data subset size to 700 for WikiMIA and ArxivTection, and 4000 for BookMIA. Then, we run each dataset five times for robust evaluation, each with differently sampled subsets. For methods that require fine-tuning (*e.g* our Kernel Divergence Score and Fine-tuned Score Deviation (Zhang et al., 2025)), we train the model for 1 epoch using stochastic gradient descent with a batch size of 4. Furthermore, in determining the bandwidth parameter $\gamma$ in the RBF kernel (Eq. (2)), we utilize the median heuristic (Garreau et al., 2017). Further implementation details are in Appendix A.

### 4.2. Experimental Results

**Kernel divergence score satisfies the monotonicity requirement.** To evaluate compliance with the monotonicity requirement, we analyze the correlation between scores and contamination rates $\lambda \in \{0.0, 0.05, 0.10, \dots, 0.95, 1.0\}$. The primary metric used is the Spearman correlation coefficient, which directly evaluates the monotonic relationship between scores and contamination rates, ensuring alignment with the expected trend. Additionally, we compute the Pearson correlation coefficient to provide insight into the linearity of the trends, with higher values indicating a stronger linear pattern in the scores.

In Table 1, we present the correlation coefficients for the three benchmark datasets. We keep the dataset size fixed while varying the contamination ratios. We observe that existing approaches often exhibit highly varying correlation and even reversed signs. For instance, in the BookMIA dataset, our thorough evaluation across five random subsets consistently revealed negative correlation values for several baseline methods. In contrast, KDS consistently achieves a near-perfect correlation on all datasets. On average, it demonstrates the strongest compliance with the monotonicity requirement.

Moreover, it is noteworthy that compared to FSD (Zhang et al., 2025) in Table 1, which also leverages fine-tuning information in detecting pre-training data, our method demonstrates more consistent performance improvements. We attribute this improvement to our KDS's direct assessment of the structural information within model representations, bypassing the reliance on intermediate scoring adjustments in FSD methods. This direct approach allows our score to effectively capture the intrinsic characteristics of the data, leading to more reliable scoring.

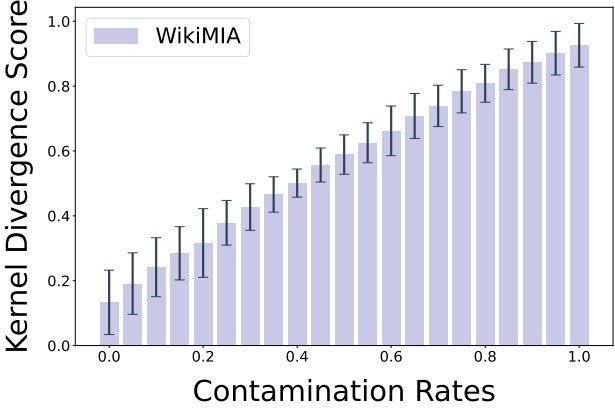

*Figure 3.* **Trend of Kernel Divergence Scores on WikiMIA.** The score shows monotonic increase with respect to contamination rate, and the standard deviation over 5 runs is low.

**Kernel divergence score satisfies the consistency requirement.** To verify the Consistency requirement, we test whether our score remains stable across independently and identically distributed datasets sampled from the same distribution and with the same contamination rate $\lambda$. For each contamination rate $\lambda \in \{0.0, 0.05, 0.10, \ldots, 0.95, 1.0\}$, we create datasets by randomly sampling 5 independent subsets from each dataset. Each subset complies with the mixing rate $\lambda$, consisting of seen and unseen samples in proportions determined by $\lambda$. All datasets are fixed to the same size to ensure comparability. In Figure 3, we observe that the kernel divergence score demonstrates relatively low standard deviations, indicating compliance with the Consistency requirement. This shows that our method can produce stable and reliable scores, independent of the specific random subset used.

## 5. Ablation Studies

In this section, we conduct an in-depth analysis to understand various design choices of our kernel divergence score.

**What's the impact of fine-tuning on the score?** As described in Eq. (4), the Kernel Divergence Score comprises two key components: **(1)** the kernel similarity matrix $\Phi(Z)$, which serves as a soft gating mechanism, and **(2)** the change in pairwise distance, which captures the effects of supervised fine-tuning. The roles of these two components are qualitatively illustrated in the left and middle panels of Figure 2. To further elucidate their individual contributions, we conducted an ablation study, with results summarized in Table 2 (top). The study evaluates the impact of removing each component.

Specifically, ablating component (1) involves omitting the soft gating mechanism, thereby utilizing the average of

*Table 2.* **Ablation study of KDS components.** Correlation coefficients are averaged across 5 independent runs evaluated on the WikiMIA dataset.

| Methods | Spearman ↑ | Pearson ↑ |
|---|---|---|
| **Kernel Divergence Score** | 0.999 | 0.993 |
| w/o (1) Soft Gating | 0.998 | 0.990 |
| w/o (2) Fine-tuning | 0.683 | 0.749 |
| w/ Euclidean Distance Kernel | 0.999 | 0.994 |
| w/ Cosine Similarity Kernel | 0.998 | 0.990 |
| w/ Dot-product Kernel | 0.998 | 0.986 |

$\left| ||Z_i' - Z_j'||_2^2 - ||Z_i - Z_j||_2^2 \right|$ as the contamination score. Conversely, ablating component (2) is equivalent to disregarding the effects of supervised fine-tuning, relying solely on the kernel similarity matrix $\Phi(Z)$ before fine-tuning, with the average kernel entry value serving as the contamination score. The results indicate that removing the soft gating mechanism (Component 1) leads to a marginal decline in performance. This suggests that while the gating mechanism enhances the score's reliability, the majority of the information is derived from the differential effects observed before and after fine-tuning. Indeed, removing the supervised fine-tuning component (Component 2) significantly degrades the performance, highlighting the critical role of fine-tuning in amplifying the kernel's ability to measure dataset contamination levels.

**Our method is not sensitive to the choice of kernel function.** The Kernel Divergence Score employs the RBF kernel to compute differences after supervised fine-tuning. However, an important question arises: how robust is the scoring performance to variations in the kernel function? To address this, we evaluate our method using alternative kernel functions, including Euclidean pairwise distance, cosine similarity, and dot-product similarity, as shown in Table 2 (bottom). For the Euclidean pairwise distance and cosine similarity kernels, we replace $\Phi(Z)$ in Eq. (5) with the respective kernel computations. We add 1 to the cosine similarity kernel matrix to enforce non-negative entries. For the dot-product similarity kernel, we compute the mean squared error of the kernel matrices before and after fine-tuning, as the kernel entries may take negative values, making them incompatible with the logarithmic operation used in our scoring function.

The results indicate that scoring performance remains consistent across different kernel functions. This suggests that the effectiveness of the Kernel Divergence Score lies in its ability to leverage structural information from kernel representations, rather than being dependent on the specific choice of the kernel. These findings highlight the versatility of our kernel-based approach in measuring contamination levels across diverse settings.

*Table 3.* **Effect of different kernel bandwidths.** Scoring performance metrics are retrieved and averaged over 5 independent runs.

| Kernel Bandwidth | Spearman ↑ | Pearson ↑ |
|---|---|---|
| $\gamma$ = Median | 0.999 | 0.993 |
| $\gamma$ = 0.001 | 0.999 | 0.994 |
| $\gamma$ = 0.01 | 0.999 | 0.994 |
| $\gamma$ = 0.1 | 0.999 | 0.994 |
| $\gamma$ = 1.0 | 0.999 | 0.994 |
| $\gamma$ = 10.0 | 0.801 | 0.838 |

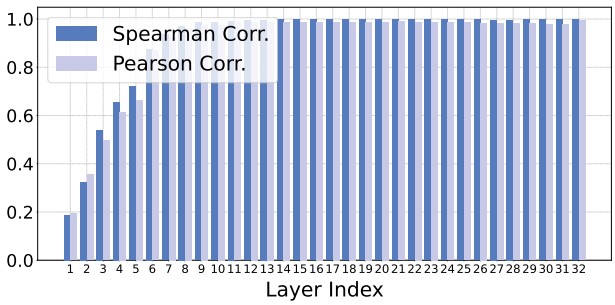

*Figure 4.* **Scoring performance across embedding location.** Correlation coefficients from different layers are retrieved using Mistral-7B-Instruct-v0.2 on WikiMIA.

**How does the kernel bandwidth $\gamma$ impact the performance?** In an RBF kernel, the bandwidth parameter $\gamma$ controls the sharpness of the kernel's entry distribution. In our approach, we employed the median heuristic (Garreau et al., 2017), which sets $\gamma$ as the inverse of the median pairwise distance. To examine the effect of $\gamma$ on contamination scoring, we evaluate different bandwidth values $\{0.001, 0.01, 0.1, 1.0, 10.0\}$, as shown in Table 3. The results indicate that scoring performance is largely invariant to the choice of $\gamma$. This behavior is expected, considering that Eq. (4) with arbitrary $\gamma$ is

$$\frac{1}{E} \sum_{i,j=1}^{n} \underbrace{\gamma \exp(-u_{i,j})^{\gamma}}_{\text{monotonicity preserved}} |u'_{i,j} - u_{i,j}|, \quad (6)$$

where $u_{i,j} = ||Z_i - Z_j||_2^2$ and $u'_{i,j} = ||Z'_i - Z'_j||_2^2$. The effect of $\gamma$ is limited to a constant multiplicative factor on the overall scores, and to a power factor that controls the sharpness of the soft gate. This does not influence their relative trends across varying contamination rates. This invariance underscores the robustness of our Kernel Divergence Score to the choice of bandwidth parameter. However, setting an excessively large value (*e.g.* $\gamma = 10.0$) may cause numerical errors and degrade scoring performance.

*Table 4.* **Effect of different training configurations.** Scoring performance metrics are retrieved and averaged over 5 independent runs. Values in parentheses are the standard deviation across runs.

| Configuration | Spearman ↑ | Pearson ↑ |
|---|---|---|
| 1 Epoch; Stochastic GD | 0.999 (0.00) | 0.993 (0.00) |
| 1 Epoch; Batch GD | 0.908 (0.18) | 0.916 (0.13) |
| 4 Epochs; Stochastic GD | 0.983 (0.02) | 0.643 (0.43) |
| 4 Epochs; Batch GD | 0.846 (0.15) | 0.875 (0.08) |

**What's the impact of embedding extraction location?** In our main experiments, we use the output embeddings from the final layer of the LLM to compute KDS. To further analyze the impact of embedding location, we evaluate the Spearman and Pearson correlation using embeddings extracted from different layers of the model, as shown in Figure 4 (top). Our results reveal that the strongest correlation is observed in the latter layers, indicating that these layers contain the most information relevant to dataset contamination. This suggests that the latter layers of the LLM, which are typically fine-tuned to align with specific tasks or datasets, are more sensitive to the effects of contamination compared to earlier layers. These findings support the selection of final layer embeddings for kernel computation, as they provide an informative basis for assessing dataset contamination.

**How does SFT configuration impact the performance?** Table 4 presents the Spearman and Pearson correlation coefficients for contamination scores under different SFT training configurations on the WikiMIA dataset. The configurations vary in terms of the optimization method (Stochastic Gradient Descent vs. Batch Gradient Descent) and the number of fine-tuning epochs (1 vs. 4). The results show that stochastic GD significantly outperforms batch GD, suggesting that the finer-grained updates introduced by SGD enhance the sensitivity of the Kernel Divergence Score to dataset contamination. On the other hand, increasing the number of fine-tuning epochs does not necessarily improve scoring performance. This may be attributed to the repeated exposure of the model to the same samples during training, which could obscure the distinction between seen and unseen samples. Overall, training with one epoch using SGD leads to the best performance.

## 6. Discussions

**Temporal shift problems of MIA benchmarks.** Recent studies have expressed concerns regarding the temporal shift issues in existing Membership Inference Attack (MIA) benchmarks (Duan et al., 2024; Das et al., 2024; Maini et al., 2024). Notably, datasets such as WikiMIA, BookMIA, and ArxivTection have been identified as susceptible to temporal

Table 5. Baseline comparison on PILE data subsets.

| Methods | Wikipedia | PhilPapers | Enron | HackeerNews | Pile_CC | StackExchange | Average |
|---|---|---|---|---|---|---|---|
| Zlib | 0.861 | **1.000** | **1.000** | -0.956 | -0.782 | 0.990 | 0.352 |
| Zlib + FSD | **1.000** | 0.991 | 0.999 | 0.323 | 0.894 | 0.999 | 0.868 |
| Perplexity | -0.886 | 0.999 | 0.999 | -0.999 | -0.251 | 0.999 | 0.144 |
| Perplexity + FSD | **1.000** | 0.990 | 0.999 | 0.118 | **0.908** | **1.000** | 0.836 |
| Min-K% | -0.645 | 0.996 | **1.000** | -0.955 | 0.690 | 0.999 | 0.348 |
| Min-K% + FSD | 0.997 | 0.952 | 0.997 | 0.421 | **0.908** | **1.000** | 0.879 |
| Min-K%++ | -0.482 | 0.960 | -0.842 | 0.561 | 0.514 | 0.697 | 0.235 |
| Min-K%++ + FSD | -0.536 | 0.994 | -0.770 | 0.705 | -0.358 | 0.210 | 0.041 |
| **Kernel Divergence Score (Ours)** | 0.891 | 0.982 | **1.000** | **0.897** | 0.895 | **1.000** | **0.944** |

Table 6. **Evaluation on benchmarks with IID setup.** We evaluate the monotonicity of our kernel divergence score on six subsets from the Pile dataset (Gao et al., 2020).

| Data Subset | Spearman ↑ | Pearson ↑ |
|---|---|---|
| Wikipedia | 0.891 | 0.922 |
| PhilPapers | 0.982 | 0.974 |
| Enron | 1.000 | 0.965 |
| HackerNews | 0.897 | 0.920 |
| Pile-CC | 0.895 | 0.908 |
| StackExchange | 1.000 | 0.998 |
| **Average** | **0.944** | **0.948** |

Table 7. **Evaluation using various models.** We evaluate the Monotonicity on the WikiMIA dataset.

| Model | Spearman ↑ | Pearson ↑ |
|---|---|---|
| Mistral-7B-Instruct-v0.2 (Jiang et al., 2023) | 0.999 | 0.996 |
| Llama-3.1-8B-Instruct (Dubey et al., 2024) | 0.982 | 0.952 |
| Phi-3-small-128k-instruct (Abdin et al., 2024) | 0.892 | 0.890 |

method reliably scores contamination levels without relying on temporal shifts.

**Extension to various model families.** We extend our evaluation to diverse models to demonstrate the versatility of our approach. As presented in Table 7, our approach consistently exhibits near-perfect correlation values across all models tested. These findings underscore the robustness and applicability of our method across diverse model families.

**Computational cost.** Our kernel divergence score involves a fine-tuning step to obtain two kernel matrices, followed by the computation of our scoring function. Given a dataset with $N$ samples, the fine-tuning step operates with a time complexity of $O(c_1 \cdot N)$, where $c_1$ is a constant influenced by factors such as average sample length, batch size, and model dimension. The computation of the KDS score has complexity $O(c_2 \cdot N^2)$, due to the quadratic nature of kernel matrix operations. In practice, the latency overhead caused by scoring is minimal, as these operations are highly optimized through vectorized computations. As demonstrated in Table 8, the latency measured in seconds for each dataset confirms the efficiency and scalability of our approach.

# 7. Related Works

Data contamination (Magar & Schwartz, 2022; Xu et al., 2024; Balloccu et al., 2024; Dekoninck et al., 2024), also known as benchmark leakage, poses a significant challenge in the evaluation of LLMs (Zhou et al., 2023; Duan et al.,

cues, which can inadvertently simplify the membership inference task. This simplification arises because models can exploit temporal information to distinguish between seen versus unseen data, leading to a potential overestimation of detection performance.

To ensure the robustness of our approach and mitigate potential biases introduced by temporal shifts, we conducted evaluations using 500 samples from six subsets of the Pile dataset (Gao et al., 2020). The subsets include text data from various sources, including expository proses (Wikipedia), academic papers (PhilPapers), emails (Enron), news articles (HackerNews), web-scraped data (Pile-CC), and user-contributed questions and answers (StackExchange). For each subset, the 'train' set is regarded as seen data, while the 'val' set serves as unseen data. We mix these two sets according to varying contamination rates to assess our model's performance under different conditions. This methodology provides a rigorous assessment, ensuring that our model does not exploit temporal cues. In Table 5, we provide the baseline comparison results evaluated on the PILE data subsets. On average, our Kernel Divergence Score (KDS) achieves the highest Spearman correlation coefficient. Moreover, as presented in Table 6, both the Spearman and Pearson correlation coefficients are near 1.0, averaging at 0.944 and 0.948, respectively. These findings demonstrate that our

*Table 8.* **Computational cost.** We report the computation time in seconds, measured on a single RTX H200 GPU.

| Dataset | Size ($N$) | Fine-tuning | Scoring |
|---|---|---|---|
| Complexity | | $O(c_1 \cdot N)$ | $O(c_2 \cdot N^2)$ |
| WikiMIA | 700 | 39s | 0.0008s |
| WikiMIA | 350 | 23s | 0.0006s |
| BookMIA | 4000 | 555s | 0.0013s |
| BookMIA | 2000 | 294s | 0.0012s |
| BookMIA | 1000 | 114s | 0.0011s |

2024). To mitigate this problem, one line of research focuses on "decontaminating" datasets by introducing controlled perturbations to reduce overlap with evaluation (Yang et al., 2023). Another line explores methods for detecting contaminated datasets or identifying samples seen during LLM training. Membership inference attack (MIA) techniques (Shokri et al., 2017; Truex et al., 2019) have been employed to classify individual data as seen or unseen (Yeom et al., 2018; Salem et al., 2019; Mattern et al., 2023), with many recent studies specifically targeting LLMs for pre-training data detection (Carlini et al., 2021; Shi et al., 2023; Zhang et al., 2024b; Xie et al., 2024; Li, 2023; Ye et al., 2024). In addition, set-level detection methods have been introduced to identify contamination at a broader dataset level (Oren et al., 2024; Zhang et al., 2024a; Golchin & Surdeanu, 2024). Building on this foundation, our work introduced a novel approach, Kernel Divergence Score, to scoring contamination levels using information derived from embedding kernel similarity matrices. *An expanded literature review of MIA is in Appendix F.*

## 8. Conclusion

In this work, we addressed the critical issue of dataset contamination in LLM by introducing the Kernel Divergence Score. By capturing fine-tuning-induced shifts in sample embeddings, KDS provides a robust and interpretable measure of contamination. Extensive experiments on controlled scenarios demonstrated the effectiveness of KDS in satisfying key properties like monotonicity and consistency, outperforming existing baselines. This work paves the way for more reliable benchmark evaluations, fostering better dataset curation practices in LLM research.

## Impact Statement

The broader impact of this work lies in its potential to significantly improve the reliability, transparency, and fairness of large language model evaluation. By enabling the identification and quantification of contaminated datasets, our approach ensures that reported performance metrics are more trustworthy and reflective of a model's true generalization

capabilities. This contributes to a more rigorous benchmarking process, fostering fair and meaningful comparisons across different models and architectures. Furthermore, the insights gained from this work can inform better practices for dataset curation. This not only reduces the risk of inflated performance results but also enhances the utility of benchmarks as tools for guiding research and development.

## Limitations and Future Work

In this work, we introduced the Kernel Divergence Score as a method to reliably measure dataset contamination levels. Compared to sample-wise scoring methods, our kernel divergence score leverages kernels that require quadratic computation. As shown in Table 8, kernel computation itself is not a significant burden in practice. However, it can become a bottleneck in terms of time and space as dataset size increases substantially. To mitigate this computational challenge, we can reduce the amount of computation by adopting local kernel approaches (Segata & Blanzieri, 2010), and by sequentially computing and accumulating the kernel entries to avoid out-of-memory errors. Another promising direction for future research is the application of Positive-Unlabeled (PU) learning (Elkan & Noto, 2008), which focuses on constructing classifiers using only positive and unlabeled data. In the context of dataset contamination detection, PU learning could help in estimating the distribution of contaminated data when only a subset of positive (*i.e.*, contaminated) examples is available. Additionally, exploring kernel calibration methods may enhance the robustness of KDS across different models. By investigating these directions, future work can aim to develop a more universally applicable contamination scoring mechanism, thereby improving the reliability and comparability of contamination assessments across various machine learning models and datasets.

## Acknowledgement

The authors would like to thank Shawn Im and Seongheon Park for their valuable comments on the manuscript. Hyeong Kyu Choi, Maxim Khanov, and Yixuan Li are supported in part by the AFOSR Young Investigator Program under award number FA9550-23-1-0184, National Science Foundation under awards IIS-2237037 and IIS-2331669, Office of Naval Research under grant number N00014-23-1-2643, Schmidt Sciences Foundation, and Alfred P. Sloan Fellowship.

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

# Appendix

## Contents

## A. Further Experimental Details

In this section, we append further experimental details and provide formal definitions of the baselines evaluated in the manuscript.

### A.1. Implementation Details

For supervised fine-tuning, we utilize Low-rank Adaptation (Hu et al., 2021). In Table 9, we disclose detailed LoRA configurations and other training hyperparameters used for supervised fine-tuning.

*Table 9.* **Supervised Fine-tuning Configurations and Hyperparameters.**

| Hyperparameter | WikiMIA | BookMIA | ArxivTection |
|---|---|---|---|
| LoRA Dimension | | 8 | |
| LoRA $\alpha$ | | 32 | |
| LoRA Dropout | | 0.1 | |
| LoRA Target Modules | | query, value projection layers | |
| SGD Learning Rate | | 0.0001 | |
| Batch GD Learning Rate | | 0.01 | |
| Batch Size | | 4 | |
| Dataset Size | 700 | 4000 | 700 |

### A.2. Baseline Definitions

Here, we provide formal definitions for each baseline compared in Table 1.

**Definition 1.** (*Zlib Score*) *is the negated ratio of the log perplexity and the zlib compression size:*

$$-\frac{1}{n}\sum_{i=1}^{n}\frac{-\frac{1}{|\mathcal{T}_i|}\sum_{x_j\in\mathcal{T}_i}\log P_\theta(x_j|x_{<j})}{\text{Zlib}(\mathbf{x}_i).\text{size}}, \tag{7}$$

*where $\mathcal{T}_i$ is the set of tokens from sample $i$.* (Carlini et al., 2021)

**Definition 2.** (*Perplexity Score*) *is the negated average perplexity across samples:*

$$-\frac{1}{n}\sum_{i=1}^{n}\exp\left(-\frac{1}{|\mathcal{T}_i|}\sum_{x_j\in\mathcal{T}_i}\log P_\theta(x_j|x_{<j})\right), \tag{8}$$

*where $\mathcal{T}_i$ is the set of tokens from sample $i$.* (Li, 2023)

**Definition 3.** (*Min-K% Score*) *is the negated mean probability from bottom-$k\%$ tokens averaged across samples:*

$$-\frac{1}{n\cdot|\mathcal{K}_i|}\sum_{i=1}^{n}\sum_{x_j\in\mathcal{K}_i}\log P_\theta(x_j|x_{<j}), \tag{9}$$

*where $\mathcal{K}_i$ is the set of bottom-$k\%$ tokens from sample $i$.* (Shi et al., 2023)

**Definition 4.** (*Min-K%++ Score*) *is the negated mean normalized probability from bottom-$k\%$ tokens averaged across samples:*

$$-\frac{1}{n\cdot|\mathcal{K}_i|}\sum_{i=1}^{n}\sum_{x_j\in\mathcal{K}_i}\frac{\log P_\theta(x_j|x_{<j})-\mu_{x_{<j}}}{\sigma_{x_{<j}}}, \tag{10}$$

*where $\mathcal{K}_i$ is the set of bottom-$k\%$ tokens from sample $i$, $\mu_{x_{<j}}=\mathbb{E}_{z\sim p(\cdot|x_{<j})}[\log p(z|x_{<j})]$ is the expected log probability over the vocabulary of the model, and $\sigma_{x_{<j}}=\sqrt{\mathbb{E}_{z\sim p(\cdot|x_{<j})}[(\log p(z|x_{<j})-\mu_{x_{<j}})^2]}$ is the standard deviation.* (Zhang et al., 2024b)

Following the general guideline from Shi et al. (2023), we take the bottom 20% tokens for the Min-K% Score and Min-K%++ Score.

**Definition 5.** (*Fine-tuned Score Deviation*) *is the difference of scores before and after supervised fine-tuning, averaged across samples:*

$$\frac{1}{n}\sum_{i=1}^{n}S(\mathbf{x}_i;\theta)-S(\mathbf{x}_i;\theta'), \tag{11}$$

*where $\mathbf{x}_i$ is the $i$-th sample in the dataset, $S(\cdot;\cdot)$ is an existing scoring function (e.g., Min-K% or Perplexity Score), and $\theta,\theta'$ are models before and after fine-tuning, respectively.* (Zhang et al., 2025)

**Definition 6.** (*Sharded Rank Comparison Test*) *is the difference between the log likelihood of the canonical dataset sample ordering from the mean over shuffled sample orderings, averaged across dataset shards:*

$$\frac{1}{r}\sum_{k=1}^{r}\left[\log P([x_i^{(k)}]_{i=1}^{n})-\frac{1}{|\mathfrak{S}|}\sum_{\sigma\in\mathfrak{S}}\log P([x_{\sigma(i)}^{(k)}]_{i=1}^{n})\right], \tag{12}$$

*where $r$ is the number of shards, $\mathfrak{S}$ is the set of sample permutations, and $[x_i^{(k)}]_{i=1}^{n}$ is the sequence of samples $x_1,x_2,\ldots,x_n$ in $k$-th shard of the dataset.* (Oren et al., 2024)

## B. Kernel Decomposition Plots Across Contamination Rates

In this section, we extend Figure 2 by providing visualization of kernel components at contamination $\lambda=\{0.2,0.4,0.6,0.8\}$. In all cases, the pattern shown in kernels is consistent with our explanation in Section 3.

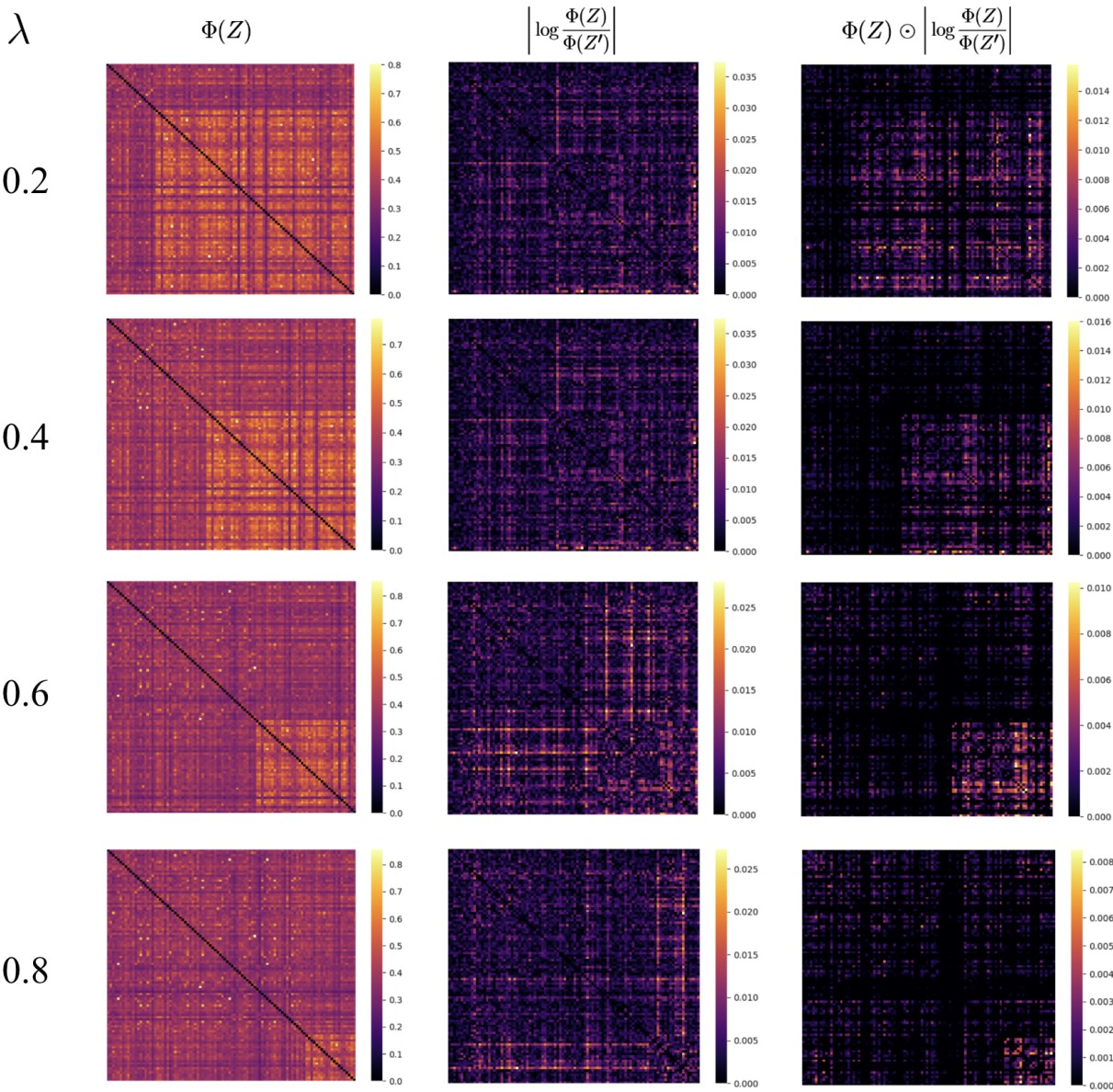

*Figure 5.* **Decomposition of the Kernel Divergence Score - full list.**

## C. Consistency Requirement Compliance in Detail

To assess the adherence of each baseline to the Consistency requirement, we calculate the Mean Absolute Percentage Error (MAPE) over five independent runs.

$$\text{MAPE} = \frac{1}{5} \sum_{t=1}^{5} \left| \frac{S_t - \bar{S}}{\bar{S}} \right|,$$

where $S_t$ represents the score from the $t$-th run, and $\bar{S}$ denotes the mean score across all five runs. This metric is then averaged across all contamination rates. The results, presented in Table 10, indicate that our Kernel Divergence Score achieves the lowest MAPE among non-FSD-based methods, highlighting its superior consistency. FSD-based scores generally have lower average MAPE, but they often fail to meet monotonicity requirements (Table 1).

*Table 10.* **Consistency Requirement in terms of average MAPE.** The average Mean Absolute Percentage Error (MAPE) over 5 independent runs is calculated. Among non-FSD baselines, our Kernel Divergence Score achieves the lowest average MAPE.

| Methods | WikiMIA | BookMIA | ArxivTection | *Average* |
|---|---|---|---|---|
| *Non-FSD-based Scores* | | | | |
| Zlib | 0.1300 | 0.1144 | 0.5199 | 0.2548 |
| Perplexity Score | 0.1786 | 0.1974 | 0.1892 | 0.1884 |
| Min-K% | 0.1966 | 0.2519 | 0.1882 | 0.2122 |
| Min-K%++ | 0.3281 | 0.1115 | 0.3268 | 0.2554 |
| *FSD-based Scores* | | | | |
| Zlib + FSD | 0.1421 | 0.1460 | 0.2191 | 0.1691 |
| Perplexity Score + FSD | 0.1587 | 0.1653 | 0.2490 | 0.1910 |
| Min-K% + FSD | 0.2936 | OOM | 0.2616 | 0.2776 |
| Min-K%++ + FSD | 0.2914 | OOM | 0.2652 | 0.2783 |
| **Kernel Divergence Score** | 0.1427 | 0.1500 | 0.2527 | 0.1818 |

## D. Role of the Normalization Factor

Recall that in our Kernel Divergence Score, we define the normalizer $E$ as the square root of the sum of entries in the kernel matrix. We employ this square-root normalizer because, despite the second-order nature of Eq. (4), the sum of kernel entries reveals a linear relationship with varying data subset sizes, as shown in Figure 6. A linear fit to the data yields an $R^2$ value of 0.9766, confirming the linearity of this relationship. Therefore, to mitigate the influence of dataset size and prevent over-penalizing the score scale, we utilize the square-root normalizer.

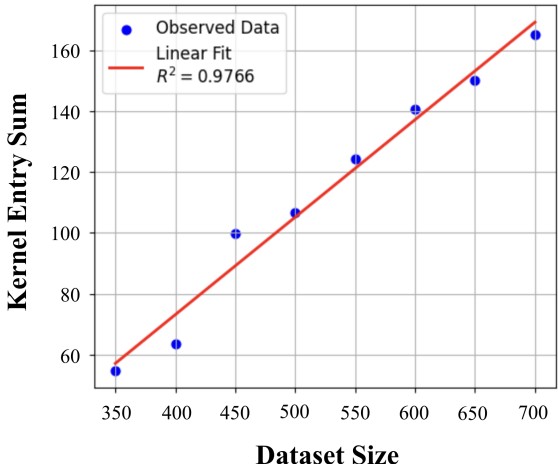

*Figure 6.* **Sum of kernel entries by data subset size.** The data subsets are randomly sampled from the WikiMIA dataset.

# E. Histogram of Change in Embedding Distances

In Figure 7, we present histograms of the change in embedding distances before and after fine-tuning. We compare three pair types: seen-seen, seen-unseen, and unseen-unseen. The figure reveals that unseen-unseen pairs exhibit more significant shifts, with distances increasing more after fine-tuning overall. This suggests that representations of unseen samples tend to spread farther apart, highlighting and confirmaing our intuition on the distributional shift induced by the fine-tuning process.

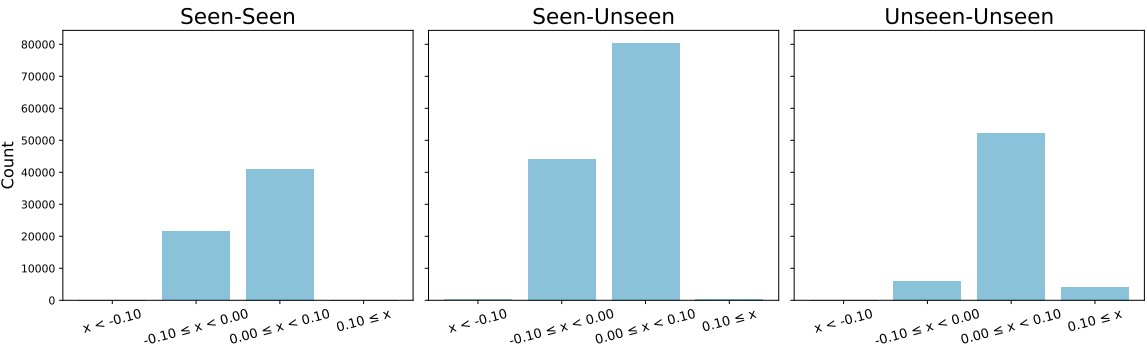

*Figure 7.* **Histogram on the values of embedding distance change.**

# F. Extended Literature Review

Here, we provide a more descriptive review of previous works on membership inference attack (MIA) (Shokri et al., 2017; Truex et al., 2019), as it is related to the objective of our work in quantifying leakage (*i.e.*, contamination) in datasets (Magar & Schwartz, 2022; Xu et al., 2024; Balloccu et al., 2024).

**Early MIA approaches.** Membership inference attacks aim to determine whether a specific data sample was part of a model's training dataset. Early approaches primarily utilized metrics derived from model outputs to achieve this. For instance, Salem et al. (2019) employed output entropy as a distinguishing factor, observing that models often exhibit lower entropy (*i.e.*, higher confidence) for training samples compared to non-training ones. Similarly, Liu et al. (2019) focused on the model's confidence scores, noting that higher confidence levels could indicate a sample's presence in the training set, and Carlini et al. (2022) proposed a likelihood ratio-based approach. Beyond output-based metrics, researchers have explored the impact of training dynamics on MIAs. Yeom et al. (2018) investigated the use of loss values, finding that models tend to produce lower loss for training samples, making loss a viable metric for membership inference. Additionally, Liu et al. (2023) introduced a gradient-based approach, leveraging the observation that the gradients of training samples differ from those of non-training samples, thereby serving as an effective indicator for membership inference.

**Challenges of MIA on Large Language Models.** While MIAs have been effective against traditional machine learning models, applying them to large language models (LLMs) presents unique challenges. Recent studies have highlighted the difficulty of performing MIAs on LLMs. For instance, Duan et al. (2024) found that MIAs often barely outperform random guessing when applied to LLMs. This limited success is primarily due to the vast scale of LLM training datasets and the relatively brief exposure of each sample during training–typically only one epoch–resulting in minimal memorization of individual data points. Additionally, the inherent overlap of n-grams between seen and unseen text samples complicates the distinction between seen and unseen data. This overlap creates a fuzzy boundary, making it challenging for MIAs to accurately infer membership. Furthermore, Meeus et al. (2024) identified that distribution shifts between collected member and non-member datasets can lead to misleading MIA performance evaluations. They demonstrated that significant distribution shifts might cause high MIA performance, not due to actual memorization by the model, but because of these shifts. When controlling for such shifts, MIA performance often drops to near-random levels.

**MIA on Large Language Models.** Despite challenges, numerous studies have endeavored to apply membership inference attacks (MIA) for large language models (LLMs). Building on classical appraoches (Yeom et al., 2018; Carlini et al., 2022), researchers have introduced a range of innovative approaches tailored to LLMs. Perplexity-based methods have been utilized, as demonstrated by Li (2023) and Carlini et al. (2021), who leveraged perplexity as a key metric to infer membership. Similarly, likelihood-based strategies have been explored, with Shi et al. (2023) and Zhang et al. (2024b) employing likelihood scores to distinguish between seen and unseen samples effectively. Other studies have extended traditional metric-based approaches to the LLM domain (Duan et al., 2024; Xie et al., 2024; Zhang et al., 2024b; Mattern et al., 2023; Ye et al., 2024), while Zhang et al. (2025) further expanded the scope by investigating the influence of fine-tuning on various membership-related scores. In addition to individual sample-level techniques, set-level detection methods have been introduced to identify contamination across broader datasets. For example, Oren et al. (2024), Zhang et al. (2024a), and Golchin & Surdeanu (2024) enabled a more holistic assessment of dataset contamination. Building on these foundations, our work introduced the Kernel Divergence Score, a novel method for evaluating contamination levels. This approach capitalizes on the differential effect of fine-tuning on embedding kernel similarity matrices, offering a unique perspective on contamination detection and addressing key limitations of existing methods.

