# OpenReview forum: "How Contaminated Is Your Benchmark? Measuring Dataset Leakage in Large Language Models with Kernel Divergence"
_ICML.cc/2025/Conference — ICML 2025 poster_

### Official Review · Reviewer_5cjP · 2025-02-18

**Overall Recommendation:** 4

**Summary:**

The paper proposes the Kernel Divergence Score to estimate data contamination in large language models. The method makes use of a kernel with layer embeddings to estimate the similarities between samples before and after finetuning. If these embeddings remain similar, the data is likely contaminated. The authors perform experiments on several benchmarks showcasing the effectiveness of their approach.

**Claims And Evidence:**

Overall, the claims made by the authors are supported by convincing evidence. However, several major concerns (see experimental designs) limit the conclusions that can be drawn from the numbers shown. Furthermore, I am particularly concerned with the very small gain compared to the Perplexity method. This baseline is obviously flawed, as lower perplexity can have any number of causes. Yet, it performs almost perfect on the given benchmark and is only outperformed by 2% by the new method. This indicates very marginal gains over a very basic baseline and therefore does not warrant the additional complexity of the method. Furthermore, the high deviations reported in Table 9 (>16%) indicate that these differences are likely not even significant.

**Essential References Not Discussed:**

N/A

**Experimental Designs Or Analyses:**

- I am concerned with the main metric used for evaluation (spearman/peason) correlation. In practice, one cannot perform such an experiment: the authors should be provide a clear cutoff at which a benchmark should be considered contaminated, or give an indication which value of their score function corresponds to what level of contamination. Without such a correspondence between the scoring function and contamination, the test has no practical use and cannot be considered a contribution. Thus, I ask the authors to investigate the following aspects:
  - Do similar values of S for the different benchmarks correspond to the same level of contamination? If not, then no such correspondence exist.
  -  Do similar values of S for different models correspond to the same level of contamination? If not, then no such correspondence exist.
- I am furthermore concerned that the test does not measure contamination, but difficulty. Using a similar argument as the authors, one could argue that training on more difficult samples the model is less used to, would cause more deviation on the sample. The fact that the perplexity metric (another metric for which this can be argued) is so close in performance to KDS indicates as well that this might be a problem. Therefore, I urge the authors to:
  - On the benchmarks presented in appendix E, show the correlation between benchmark difficulty and model score for each model separately. I suspect (but am not sure) the correlation will be significantly above 0.
  - The authors should clearly argue why this is not the case. If one wants to use this test in practical scenarios to find contamination and accuse model providers that their model is contaminated, there should be no argument what the test is measuring.
- I am concerned that the shown plots indicate a discrepancy between seen and unseen samples. Specifically, on the left-hand side of each plot, one would expect the square corresponding to seen samples to be as bright as unseen samples: brightness indicates similarity between the samples. It seems here that the unseen samples are very similar while the seen samples are not. The authors should clarify if this is a mistaken interpretation on my side (if so, explain), and if not, see how this can affect their scores as it greatly decreases the difficulty for their method to detect unseen samples.
- I am concerned with the negative correlations shown for some baselines. The authors only briefly comment on this, but perfect negative correlation (see Zlib) for a benchmark indicates that the method would work perfect if you were to swap the sign for this benchmark. This either indicates (1) the authors have found an issue no prior work has ever been able to find (2) the implementation is wrong, or (3) the metric used is not a good metric for performance. The authors are strongly encouraged to provide evidence that excludes the second and third option, since the first one is quite a strong claim.
- The authors should compare against baselines in Table 5. Due to the cited papers (Duan et al, Das et al, ...), the numbers presented in Table 1 are a bit meaningless due to the possibility of a temporal split.
- The comparison with Oren et al. is not accurate. SRCT can only be used on the canonical ordering of samples. Since the authors subsample to report the numbers, the canonical ordering disappears. The authors should ensure that the subsampling happens in such a way that consecutive samples are included for the comparison against Oren et al.

**Methods And Evaluation Criteria:**

See experimental designs.

**Other Comments Or Suggestions:**

I am currently leaning towards rejection. If the authors are able to address my concerns, I am willing to increase my score. However, if one of the following aspects is not clarified or improved during the rebuttal, I will further reduce my score:
- Very marginal gains compared to the perplexity method.
- Correspondence between S and level of contamination.
- Is difficulty correlated with KDS?

**Other Strengths And Weaknesses:**

The paper is very well-written and easy to follow.

### update after rebuttal
I have changed my view on this paper after the rebuttal. Most of my concerns were sufficiently addressed, although some concerns still remain (practical applicability of the method being the main bottleneck). Therefore, I have raised my score to a recommended accept, but would not recommend the paper for a spotlight paper.

**Questions For Authors:**

See above.

**Relation To Broader Scientific Literature:**

The authors do a good job of relating the work to existing work in data contamination. No essential work is missing, and the baselines compared against are accurate. The first paragraph in Section 6 is necessary and appreciated (albeit not entirely complete, see Experimental Designs).

**Theoretical Claims:**

I have one concern with the current requirements that are listed: Why do the authors use two requirements? Consistency follows from monotonicity (if one replaces the "if" with an "if and only if", which would be necessary for a good definition of monotonicity). Thus, Requirement 2 is redundant and also plays a much smaller role in the evaluation.

---

> ### Author Rebuttal · Authors · 2025-03-30
>
> We sincerely thank the reviewer for the thorough and constructive feedback. Below, we address the concerns in detail.
>
> ---
> _A1. Baseline comparison in Table 5 (Pile dataset)_
>
> We provide the comparison below. KDS achieves the highest average correlation.
>
> | Spearman Corr. | Wikipedia | PhilPapers | Enron | HackeerNews | Pile_CC | StackExchange | **Average** |
> |---|:---:|:---:|:---:|:---:|:---:|:---:|:---:|
> | Zlib | 0.861 | **1.000** | **1.000** | -0.956 | -0.782 | 0.990 | 0.352 |
> | Zlib + FSD | **1.000** | 0.991 | 0.999 | 0.323 | 0.894 | 0.999 | 0.868 |
> | Perplexity | -0.886 | 0.999 | 0.999 | -0.999 | -0.251 | 0.999 | 0.144 |
> | Perplexity + FSD | **1.000** | 0.990 | 0.999 | 0.118 | **0.908** | **1.000** | 0.836 |
> | Min-K% | -0.645 | 0.996 | **1.000** | -0.955 | 0.690 | 0.999 | 0.348 |
> | Min-K% + FSD | 0.997 | 0.952 | 0.997 | 0.421 | **0.908** | **1.000** | 0.879 |
> | Min-K%++ | -0.482 | 0.960 | -0.842 | 0.561 | 0.514 | 0.697 | 0.235 |
> | Min-K%++ + FSD | -0.536 | 0.994 | -0.770 | 0.705 | -0.358 | 0.210 | 0.041 |
> | KDS (Ours) | 0.891 | 0.982 | **1.000** | **0.897** | 0.895 | **1.000** | **0.944** |
>
>
> _A2. Significance over Perplexity_
>
> In the A1 table above, **KDS outperforms Perplexity on the Pile dataset, with an average correlation of 0.944 compared to 0.144**.
>
>
> _A3. Negative correlations_
>
> We resonate with the reviewer’s concern. To address the potential causes, **we reviewed our implementation and found no errors**. Since our code directly builds on established prior work, an implementation error is unlikely (code will be released).
>
> ---
> _A4. Main metrics and Practical utility_
>
> (1) Main metrics
>
> We agree that absolute contamination estimation and exact correspondence would be valuable. However, a **necessary step toward this goal** is to first validate the correctness of any scoring functions. The use of Spearman/Pearson correlation in our controlled experiments allows us to rigorously assess whether KDS (and other approaches) produces consistent and monotonic metrics, necessary conditions for any meaningful scoring function. **This provides dev-purpose metric to ensure score behaves predictably and reliably in principle, which is essential for deploying it in practical settings**. To our knowledge, no prior work has investigated this practically important angle.
>
> Concerningly, we discovered existing scores do not satisfy these necessary conditions. In contrast, KDS consistently satisfies these conditions under varying benchmarks (including the challenging ones), providing stronger foundations for future method design.
>
> (2) Practical utility
>
> The lack of access to pre-training corpora makes ground-truth estimation fundamentally challenging. Moreover, pre-training data is often non-overlapping across models, further complicating any cross-model normalization or comparison. _To our knowledge, there is no existing work that has effectively estbalished correspondence between the scoring function and dataset contamination_. **This reflects the intrinsic difficulty of the problem, rather than a unique shortcoming of our approach**.
>
> That said, we would like to emphasize that **KDS is highly practical, which supports real-world applications such as safety auditing and benchmark selection**, where knowing which dataset is more contaminated is often more important. It enables dataset creators or auditors to prioritize and select the least contaminated benchmarks for evaluation purposes. We state the practical utility in **Lines 108-117**.
>
> ---
> _A5. Correlation between benchmark difficulty and KDS_
>
> Thank you for this insightful question. During rebuttal, we computed the correlation of {Llama3.1-8b, Mistral-7b, Qwen2.5-7b}'s performance on {GSM, MPP, MPL, TFQA} with KDS. The correlation values are 0.2, 0.4, 0.2, respectively, which are all very low. This indicates that **benchmark difficulty is not what determines the scale of KDS.**
>
> ---
> _A6. Why do we need Requirement 2?_
>
> While the two requirements could be combined, separating them into orthogonal objectives offers clearer guidance for future research by isolating distinct performance aspects.
>
> ---
> _A7. Clarification on Figure 2 (left)_
>
> It is true that unseen samples are similar to each other in Figure 2 (left), and we would like to clarify that this is the similarity _before_ fine-tuning. However, **our method relies on the changes in the similarity or distance before and after fine-tuning, which is much more significant for unseen examples**. As shown in Figure 2 (middle, without gating) and Figure 2 (right, with gating), unseen samples exhibit significantly larger shifts in embedding geometry, whereas seen samples remain relatively stable. This differential behavior is the core signal that KDS captures.
>
> Ps. The performance of only using Figure 2 (left) is in Table 2 "w/o (2) Fine-tuning".
>
> ---
> _A8. Is comparison with SRCT accurate?_
> **We preserved the canonical ordering of samples in SRCT** by not shuffling the dataset.

---

> > ### Comment · Reviewer_5cjP · 2025-04-02
> >
> > Thank you for the thorough reply. Their are a couple of concerns left that I would like to see resolved before updating my review. Please see all details below.
> >
> > - A1. Baseline comparison in Table 5 (Pile dataset): Thank your for this comparison. I believe this provides much stronger evidence for your method.
> > - A2. Significance over Perplexity: Your point is accepted.
> > - A3. Negative correlations: While your point is acknowledged, I still find this extremely concerning (probably not for this paper, but more for the field in general). This is an incredibly concerning observation, and I would encourage the authors to investigate it more thoroughly: what is causing the extremely negative correlations? Is there any way to tell based on the benchmark when a method will perform worse than random?
> > - A4: I do agree that most papers do not provide such a correspondence, but there are a few: SRCT and [1]. Furthermore, the fact that other papers do not provide such a correspondence, does not mean that follow-up work should try to strive further and obtain it. Especially since it is impossible to use your method without such a correspondence in actual use cases. Furthermore, "where knowing which dataset is more contaminated is often more important" also requires the scores to match across benchmarks, which the authors have not shown. Indeed, suppose I obtain a score of 0.5 on MMLU and 0.8 on ARC-Challenge. Due to non-existence of a proper corresponde, there is no way of saying that ARC-Challenge is actually more contaminated than MMLU (and whether either is contaminated at all). Therefore, the authors have failed to address this point and I strongly question the practical effect this paper has.
> > - A5: Thank you for computing this. Is there any reason why you did not do this for all models? It seems like a pretty simple computation. Furthermore, I would definitely not say that 0.2,0.4,0.2 is very low, but it is also not very high. My concern remains, especially since the authors have not provided a reason why my argument for correlation with difficulty is incorrect or mistaken.
> > - A6: I personally disagree, as I think it would simplify the presentation if you had only one requirement. However, I acknowledge that this is a matter of personal opinion and will therefore not further push the point.
> > - A7: Yes, the authors confirm my exact concern: if the distribution of seen samples is much more concentrated then the unseen samples, would the problem of contamination detection not become easier? Ideally, the distribution of both seen and unseen samples before training is the same. Otherwise, training on one seen sample might also increase your score on another because they are so similar.
> > - A8: Great, this resolves my concern.
> > [1] https://arxiv.org/abs/2405.16281

---

> > > ### Author Response · Authors · 2025-04-03
> > >
> > > Thank you for the thorough read of our rebuttal, and probing with insightful questions. We address the remainder ones below.
> > >
> > > ---
> > > _A3. Reason for negative correlation_
> > >
> > > We fully agree with the reviewer's view. Accordingly, we have investigated this in more depth. As a case study, we examined the Zlib baseline on the HackerNews benchmark, with a correlation of -0.956. We hypothesize that these negative correlations may stem from the following factors:
> > >
> > > - Many baseline methods operate at the instance level and then aggregate scores across a dataset. We observed that the overall score distribution appears to be highly similar between small vs. large contamination rate $\lambda$, making the aggregated score relatively insensitive to gradual contamination. In such cases, even a few outliers can disproportionally influence the overall score and introduce non-monotonic behavior.
> > > - Upon closer examination, we found that unseen examples can contain a large number of code snippets with predictable formatting, repetitive syntax and tokens, which makes them highly compressible. As a result, even at a low contamination rate $\lambda=0.1$, the Zlib score can be slightly higher than $\lambda=0.9$ (compounded with the outlier effect).
> > >
> > > We believe our results can inspire follow-up works to carefully re-examine widely used contamination detection methods, which may be worth an extensive study on its own. We thank the reviewer again for the comment. We will make this discussion more prominent in the revised version.
> > >
> > > ---
> > >
> > > _A4. Score and contamination rate_
> > >
> > > We report below the scores across different benchmarks and different contamination rates. While a perfect correspondence between score and contamination rate remains difficult to establish, we observe that the overall score trends are consistently monotonic across all three benchmarks.
> > >
> > > | **Contamination Rate** | **WikiMIA** | **Arxivtection** | **BookMIA** |
> > > |:---:|:---:|:---:|:---:|
> > > | 0.10 | 0.110 | 0.172 | 0.158 |
> > > | 0.20 | 0.228 | 0.300 | 0.302 |
> > > | 0.30 | 0.382 | 0.437 | 0.425 |
> > > | 0.40 | 0.441 | 0.496 | 0.511 |
> > > | 0.50 | 0.487 | 0.517 | 0.619 |
> > > | 0.60 | 0.597 | 0.638 | 0.722 |
> > > | 0.70 | 0.722 | 0.768 | 0.763 |
> > > | 0.80 | 0.809 | 0.863 | 0.802 |
> > > | 0.90 | 0.920 | 0.918 | 0.832 |
> > >
> > > ---
> > >
> > > _A5. Correlation between benchmark difficulty and KDS_
> > >
> > > We should have used the wording "relatively low" in our previous response. As suggested, we also computed the correlation for more models, and observed consistent findings (e.g. Llama3.2-1b and Qwen2.5-1.5b having weak correlation). We couldn't afford to run these many inferences (number of benchmarks $\times$ models) in such a short time window, given the limited compute we have to serialize the inference runs.
> > >
> > > We would like to clarify that difficulty vs. contamination are distinct notions. A benchmark’s difficulty reflects how challenging its questions or text are for human (or model) reasoning. In contrast, contamination reflects whether the exact or near-verbatim content of that benchmark was present in the model’s pretraining corpus.
> > >
> > > > "_Using a similar argument as the authors, one could argue that training on more difficult samples the model is less used to, would cause more deviation on the sample._"
> > >
> > > This does not necessarily hold. **A difficult sample may still cause minimal deviation if the model has already seen similar content during pretraining**. Sometimes, complex or niche material—such as advanced math problems or scientific text—can appear on public educational sites, making it possible for a model to memorize them during training. In such cases, despite the sample’s difficulty, the model is already well-prepared to handle it, and fine-tuning will make only minor adjustments.
> > >
> > > Conversely, even a simple sample can cause a large deviation if it is truly novel to the model—e.g., a common-sense sentence phrased in an unfamiliar way or from an novel domain. What matters is not whether the sample is inherently hard, but whether it introduces a new learning signal. Our method captures this signal by measuring representation shift induced by fine-tuning, which we argue is a more direct proxy for novelty (and hence contamination) than for difficulty.
> > >
> > > ---
> > >
> > > _A7. Similarity plot_
> > >
> > > We now understand your concern. Yes -- detection would become easier when one distribution becomes more concentrated. However, **we would like to clarify that Figure 2 was constructed primarily for clear illustration purposes**. We intentionally chose a non-i.i.d. configuration to make the effect of fine-tuning on unseen examples more visually prominent, particularly for demonstrating how our method captures representation shifts for those samples.
> > >
> > > We would like to emphasize that **all of our quantitative results and benchmark evaluations in the main paper are based on the standard i.i.d. setup**, where seen and unseen samples are drawn from the same distribution. This ensures that contamination detection is evaluated under more realistic conditions.

---

### Official Review · Reviewer_9vAw · 2025-03-09

**Overall Recommendation:** 3

**Summary:**

This paper proposes the Kernel Divergence Score (KDS) as a measure of dataset-level benchmark contamination for LLMs. KDS is computed as the weighted average of pairwise embedding vector changes across finetuning the model under investigation on the benchmark test dataset. KDS is shown to be robust to many design choices, exhibit a high degree of 'monotonicity', i.e. correlation of contamination degree and score value, and consistency, i.e., similar contamination induces similar scores.

**Claims And Evidence:**

Key claims:
* The proposed KDS has higher(er than baselines) correlation with contamination degree => substantiated with Table 1, although Min-K% performs better on 2/3 benchmarks without requiring weigh-access
* The proposed KDS has high consistency, i.e., the same contamination level leads to similar scores => Moderately well substantiated in Figure 3. Standard deviation is very large compared to score value, at low contamination rates. I.e. while Scores are not statistically significantly different for the same contamination rates, they will also not be statistically significantly different for moderate differences in contamination rate.

**Essential References Not Discussed:**

An important work on quantifying dataset-level contamination (Dekoninck et al "Constat: Performance-based contamination detection in large language models." NeurIPS'24) is neither referenced nor compared to.

**Experimental Designs Or Analyses:**

* The experimental design is sound and consistent with the claims being made.
* Note that some claims which I believe would be important for practical relevance (model comparison and establishing absolute instead of relative contamination) are not made and not analyzed.

**Methods And Evaluation Criteria:**

* The considered datasets are a reasonable choice, which could be extended with experiments investigating contamination introduced in a fine-tuning step, but this is purely optional.
* The proposed evaluation is suitable to establish the utility of KDS as a metric to rank datasets based on their contamination
* However, no effort is made to show that it permits the comparison of models (note this is not claimed as an objective)
* Similarly, no effort is made to show that it permits to show whether or not a specific model is contaminated with a specific dataset (note this is also not claimed as an objective)
* A range of ablation studies is conducted, showing the (lack of) sensitivity of KDS to many design choices

**Other Comments Or Suggestions:**

* The use of LORA should be mentioned in the main text
* Boldening only in the average in Table 1. is misleading and hides the fact that Min-K% outperfroms KDS in 2/3 datasets.
* Figure 2 mid mostly shows that seen-seen change less. Can you create a histogram of values for seen-seen, seen-unseen, unseen-unseen (without taking the absolute value preferably)?
## Questions For Authors

**Other Strengths And Weaknesses:**

### Strengths
* LLM evaluation is a topic of crucial importance and benchmark contamination is an important factor.
* KDS is conceptually relatively simple (with an even simpler version performing just as well (differences in Euclidean distance -- no soft-gating))
* KDS shows strong monotonicity across a broad range of settings and models

### Weaknesses
* Need for Finetuning the model under investigation makes the proposed approach applicable only to open-weight models
* KDS only permits a ranking of datasets by their contamination for a given model, no comparison between models or absolute statements about the contamination of a model, severely limiting its practical utility.
* Performance improvement over more broadly applicable Min-K% seems practically irrelevant.

**Questions For Authors:**

* What is the minimal contamination rate required to show it is >0 with a confidence of 95% and 99%? (This should be computable using the results visualised in Figure 3)
* How do scores correlate with contamination across different models?
* Can you conduct an experiment, Finetuning with rephrased "seen" data, simulating a setting where the model was trained on rephrased benchmark data?
* Why take the absolute value of change in embedding similarity? The provided intuition indicates that similarities should increase for previously unseen data.
* Can you conduct an experiment showing the effectiveness of your approach detecting contamination introduced during fine-tuning? (Gradient flow only from answers and data mixed with some background data.)

**Relation To Broader Scientific Literature:**

Given the objectives of the proposed method, a reasonable set of baselines is compared to, which are carefully discussed in the appendix. KDS shows strong performance across all three considered benchmarks. However, Min-K% performs better on 2/3 benchmarks, without requiring open-weights / a fine-tuning step with the difference in performance most likely being practically irrelevant (similar or worse performance of their own method in Table 5 and 6 is considered "near-perfect").

**Theoretical Claims:**

Equivalence between (3) and (4) for $\gamma=1$ is the only claim and it is correct.

---

> ### Author Rebuttal · Authors · 2025-03-30
>
> We thank the reviewer for the thoughtful and constructive feedback. Below, we address your concerns in detail.
>
> ---
> _A1. Related work_
>
> We thank the reviewer for pointing out the work by Dekoninck et al. [1], which appears to be highly relevant! It's our oversight at the time of submission and will make sure to discuss it in the updated version.
>
> [1] Dekoninck et al "Constat: Performance-based contamination detection in LLMs." NeurIPS 2024.
>
> ---
> _A2. Open weight model_
>
> We acknowledge that KDS currently requires open-weight models. This is a fair point, and we will clearly note this limitation in the paper. However, we'd like to emphasize that focusing on open-weight models is valuable for academic research because they provide transparency **essential for developing a deeper understanding of how contamination manifests inside LLMs**. In future works, adapter-based probing techniques can be explored to extend its principles to proprietary models.
>
> ---
> _A3. Significance w.r.t. min-K%_
>
> While Min-K% works well on easy benchmarks such as WikiMIA and BookMIA, its performance degrades significantly on more challenging datasets. For example, **in the three hardest PILE subsets in Table 5, Min-K% has significantly lower performance than KDS**, as shown below (numbers are Spearman correlation):
>
> | PILE | Min-K% | KDS|
> |-|-|-|
> | Wikipedia |-0.645 | **0.891** |
> | Hackernews | -0.955 | **0.897** |
> | Pile-CC | 0.690 | **0.895** |
>
> ---
> _A4. Practical utility_
>
> We agree that absolute contamination estimation is valuable. However, **a necessary step toward this goal** is to ensure that any proposed contamination scoring function satisfies core correctness criteria—namely, monotonicity and consistency with ground-truth contamination levels in controlled settings. **These conditions ensure that the score behaves predictably and reliably in principle, which is essential before deploying it in practical settings**. To our knowledge, no prior work has investigated this practically important angle.
>
> Concerningly, we discovered existing scores do not satisfy these necessary conditions. In contrast, KDS consistently satisfies these conditions under varying benchmarks (including the challenging ones), providing stronger foundations for future method design.
>
> Lastly, we would like to emphasize that KDS is also highly practical, because it supports real-world applications such as safety auditing and benchmark selection, where knowing which dataset is more contaminated is often more important. **To further demonstrate the practical utility, we include an in-the-wild evaluation across 11 public benchmarks (Appendix E)**. For these reasons, we believe KDS offers a meaningful and practical step forward for the field.
>
> ---
> _A5. Minimal contamination rate to be regarded as "significantly contaminated"_
>
> The required rate to be considered "significantly contaminated" is **0.10 (p=0.014)** under significance level $\alpha=0.05$, and **0.15 (p=0.004)** under $\alpha=0.01$.
>
> ---
> _A6. Histogram on the values of embedding similarity change_
>
> Since plots cannot be updated or shown at this stage, we provide the histogram as a frequency table below. The table is retrieved from a contamination rate of 0.5.
>
> | bins | Seen-Seen | Seen-Unseen | Unseen-Unseen |
> |:-:|:-:|:-:|:-:|
> | x < -0.10 | 0 | 202 | 22 |
> | -0.10 <= x < 0.00 | 21529 | 44112 | 6030 |
> | 0.00 <= x < 0.10 | 40953 | 80338 | 52270 |
> | 0.10 <= x | 18 | 348 | 4178 |
> | **total** | 62500 | 125000 | 62500 |
>
> _A7. Embedding similarities should increase for unseen data. Why take the absolute change?_
>
> We would like to clarify that the embedding distances increase (i.e., embedding similarities decreases) for previously unseen data after fine-tuning. This is also evidenced in the table above, where $\log \frac{\Phi(Z)}{\Phi(Z')} = ||Z'_i-Z'_j||^2_2 - ||Z_i-Z_j||^2_2$ is mostly positive for unseen-unseen pairs.
>
> We take the absolute value of the change to ensure that we capture the magnitude of deviation, regardless of direction. This makes the score robust to mixed behavior and focuses on detecting how much the geometry shifts. We will clarify this in the final version.
>
> ---
> _A8. Effect of Fine-tuning and Rephrased Data_
>
> Following the suggestion, we fine-tuned "unseen" samples from WikiMIA and Arxivtection, observing KDS increases of 22% for WikiMIA, and 11% for Arxivtection. This shows our scoring approach's ability to effectively capture contamination introduced by fine-tuning.
>
> Additionally, we experimented on WikiMIA by mixing unseen data with rephrased "seen" data at varying proportions, yielding a Spearman correlation of -0.791. This outcome is expected, as rephrasing likely blurs the distinction between seen and unseen samples.
>
> ---
> _A9. Other suggestions_
>
> Thank you for all the constructive suggestions! We will incorporate those comments, to (1) mention the use of LoRA in the main paper, (2) bold-face all the best performances in Table 1, and (3) add a histogram based on our response, A6.

---

> > ### Comment · Reviewer_9vAw · 2025-04-01
> >
> > I want to thank the Authors for their detailed response, which has clarified most of my questions. I have adjusted my score accordingly.
> >
> > I would still be curious about the cross-model correlation of KDS with contamination, even if it turns out to be low, and would like to encourage the authors to mention the susceptibility of their approach to simple rephrasing evasion.

---

> > > ### Author Response · Authors · 2025-04-03
> > >
> > > We thank the reviewer for adjusting the score. We observe that correlations across models are generally high—for instance, (bloomz, phi3-small) = 0.99, (phi3-small, qwen2.5-14b) = 0.98, and (llama3.1-8b, qwen2.5-7b) = 0.86. We will also ensure that the final version includes a discussion on rephrasing-based evasion.
> > >
> > > Thanks again for your support!

---

### Official Review · Reviewer_CbNy · 2025-03-09

**Overall Recommendation:** 3

**Summary:**

The paper introduces the Kernel Divergence Score (KDS), a method to quantify dataset contamination in LLMs by measuring changes in kernel similarity matrices of sample embeddings before and after fine-tuning.

**Claims And Evidence:**

The central claim—that KDS effectively quantifies contamination—is supported by experiments where contamination is simulated by mixing pre-training "seen" data with "unseen" data and fine-tuning. However, a critical limitation is that contamination in real LLMs occurs during pre-training, not via post-hoc fine-tuning. The experiments validate KDS in a synthetic setup but lack evidence that it generalizes to pre-training leakage. The correlation results are convincing for the tested scenarios, but the claim’s broader applicability remains unproven.

**Essential References Not Discussed:**

None.

**Experimental Designs Or Analyses:**

The controlled experiments systematically vary contamination ratios, but the contamination is artificially introduced via fine-tuning, not pre-training. This design choice raises concerns about ecological validity. For instance, real contamination involves memorization during pre-training, which may not manifest the same embedding shifts as fine-tuning. The ablation studies (e.g., kernel function choices) are thorough but do not address this core issue.

**Methods And Evaluation Criteria:**

The kernel-based approach is novel, leveraging embedding shifts post-fine-tuning. However, evaluating contamination via fine-tuning conflates two distinct processes: pre-training data leakage and adaptation during fine-tuning. The benchmarks (e.g., WikiMIA) are standard, but their use of fine-tuning to inject contamination may not reflect real-world pre-training leakage. The evaluation criteria (monotonicity, consistency) are appropriate but limited to the authors’ experimental setup.

**Other Comments Or Suggestions:**

None.

**Other Strengths And Weaknesses:**

The methodology of this paper is interesting, but the experimental part is really hard to convince me yet.

**Questions For Authors:**

1. How does KDS perform when contamination originates from pre-training data rather than fine-tuning? If experiments only use fine-tuning to simulate contamination, the method’s applicability to real-world LLM evaluation (where contamination occurs during pre-training) is unclear. Demonstrating KDS on pre-training leakage would strengthen the claims.

2. Can KDS distinguish between memorization during pre-training vs. adaptation during fine-tuning? The current setup assumes these are equivalent, but they may involve different mechanisms. A response clarifying this would affect the validity of the evaluation.

3. Have the authors tested KDS on LLMs where the pre-training corpus is partially known? This could validate whether KDS correlates with ground-truth contamination in real-world settings.

**Relation To Broader Scientific Literature:**

The work connects to membership inference attacks (MIA) and dataset contamination detection (e.g., Carlini et al., 2021; Shi et al., 2023). However, prior MIA methods focus on individual sample detection, while KDS operates at the dataset level.

**Theoretical Claims:**

The paper lacks formal theoretical guarantees for KDS. The intuition that fine-tuning impacts unseen data more is plausible but not rigorously proven. The mathematical formulation (Eq. 3–5) is empirically validated but lacks theoretical analysis of why kernel divergence should correlate with contamination in general settings.

---

> ### Author Rebuttal · Authors · 2025-03-30
>
> We thank the reviewer for the thoughtful and constructive feedback. Below, we address the key concerns:
>
> ---
> _A1. Clarification on problem setting_
>
> We absolutely agree with your viewpoint that contamination in real LLMs occurs during pre-training! We'd like to clarify that **our central goal is indeed to quantify contamination originating from pre-training, not fine-tuning**. Our methodology and evaluation are designed around pre-training leakage.
>
> In particular, we ask this question (Sec 2.1):
>
> > Given an LLM $\mathcal{M}$ and a benchmark dataset $\mathcal{D}$ (e.g., WikiMIA), to what extent has this dataset been exposed during pre-training?
>
> To address this, our key idea is the following:
>
> > If the model $\mathcal{M}$ has already been pre-trained on $\mathcal{D}$, then if we further fine-tune $\mathcal{M}$ on $\mathcal{D}$, the model would have minimal changes in embeddings due to prior exposure during pre-training (vice versa).
>
> Hence, we use a lightweight fine-tuning step as a probe to reveal how much of the evaluation dataset was likely seen during pre-training. In other words, **we are not using fine-tuning to inject contamination, but only as a mechanism to probe for the presence of whether $\mathcal{D}$ is used in pre-training**. This design choice aligns with recent work (e.g., Zhang et al. 2025, FSD) that also leverages fine-tuning to surface memorized content—but our approach uniquely leverages embedding-level structural changes, enabling a more holistic dataset-level assessment.
>
> ---
> _A2. "the contamination is artificially introduced via fine-tuning, not pre-training"_
>
> We would like to clarify that **our experimental setup simulates contamination w.r.t. pre-training, which is precisely the phenomenon our method aims to quantify**. Specifically, our controlled experiments leverage datasets such as WikiMIA, with two subsets: seen/unseen in the pre-training corpus of the LLM (e.g., Mistral-7B). By mixing these pre-training seen and unseen subsets at varying proportions, we simulate datasets with known and controllable contamination ratios relative to pre-training—not fine-tuning.
>
> ---
> _A3: “Have the authors tested KDS on LLMs with partially known pre-training corpora?"_
>
> Excellent suggestion! We further evaluated on **Pythia-6.9b**, which is known to be pre-trained on the PILE dataset. The spearman correlation coefficient is **0.999** on the Enron subset.
>
> Furthermore, our evaluation includes WikiMIA, BookMIA, and ArxivTection benchmarks, which label samples that are likely to have been seen in general LLM pre-training as "seen". We explicitly simulate varying contamination ratios and show in **Table 1** that **KDS exhibits near-perfect monotonicity and consistency, outperforming all baselines across three datasets.**
>
> ---
> _A4. Theory_
>
> While our paper primarily focuses on providing a practical and robust empirical method for quantifying pre-training contamination, we agree that theoretical analysis is an important direction. Our core intuition—that fine-tuning shifts the embeddings of unseen pre-training data more significantly than seen data—is empirically grounded, as shown in Figure 2, and supported by strong monotonicity and consistency across all benchmarks (Tables 1, 5, 6). Moreover, the mathematical formulation in Eq. (3–5) provides a principled and interpretable construction of KDS, which integrates:
> - A soft gating function that emphasizes initially similar pairs, and
> - A distance shift measure capturing how much pairwise embedding geometry is altered after fine-tuning.
>
> That said, we acknowledge that a full theoretical analysis of why kernel divergence correlates with contamination remains nontrivial. Such analysis would require carefully defined assumptions about the structure of the embedding space, the dynamics of fine-tuning, and the underlying data distribution. We consider this a promising direction and plan to explore these theoretical foundations in future work.

---

> > ### Comment · Reviewer_CbNy · 2025-04-01
> >
> > Thanks for the author's response, which addressed some of my concerns.

---

### Official Review · Reviewer_J5f8 · 2025-03-14

**Overall Recommendation:** 4

**Summary:**

This paper investigates how to quantify dataset leakage in large language models. The proposed method is inspired by the fact that fine-tuning affects the embedding relationships unseen samples more significantly than those seen samples.

The authors propose Kernel Divergence Score, using kernel similarity matrix between the embeddings before and after fine-tuning to compute a score as the degree of dataset leakage. The proposed method achieves best performance against compared baselines.

Besides, the authors conduct comprehensive ablation studies to investigate the factors like kernel bandwidth and embedding location.

**Claims And Evidence:**

Yes, the claims are well supported by main experiments, ablation studies and visualization.

**Essential References Not Discussed:**

No.

**Experimental Designs Or Analyses:**

Yes, the experimental designs are make sense and consistent with previous work.

**Methods And Evaluation Criteria:**

Yes, the proposed methods make sense.

**Other Comments Or Suggestions:**

None

**Other Strengths And Weaknesses:**

See above.

**Questions For Authors:**

None

**Relation To Broader Scientific Literature:**

Yes, the proposed method is inspired by findings from previous study.

**Theoretical Claims:**

These is not too much theoretical claims.

---

> ### Author Rebuttal · Authors · 2025-03-30
>
> Dear Reviewer J5f8,
>
> We sincerely appreciate your positive feedback and the time you've dedicated to reviewing our manuscript. Your insights are invaluable to us. Please let us know if you have any further questions.

---

> > ### Comment · Reviewer_J5f8 · 2025-04-09
> >
> > More details.
> >
> > This paper's idea is inspired by the fact that fine-tuning affects the embedding relationships involving unseen samples more significantly than those involving seen samples. As far as I know, I think using this fact to quantify dataset leakage is a clever approach. (Some contamination detection methods are too naive)
> >
> > 1. The experimental results show the effectiveness of the proposed method.
> > 2. The authors conduct extensive ablation studies, which make me feel this work is solid.
> > 3. The writing is easy to understand.
> > 4. These are reasons why I give a positive score.
> >
> > Best regards,
> > Reviewer j5f8

---

### Decision · Program_Chairs · 2025-05-01

**Decision:**

Accept (poster)

**Comment:**

This paper proposes Kernel Divergence Score (KDS), a method to quantify dataset contamination in LLMs by measuring the divergence in kernel similarities of embeddings before and after fine-tuning. KDS reliably correlates with contamination levels and outperforms existing baselines across benchmarks.

After the author-reviewer discussion, all the reviewers reached a positive consensus (2 weak accept, 2 accept). Their positive comments are mostly based on solid and sound experimental results. I also agree with the reviewers.

Overall, I recommend "Accept" for this paper.